# Decreased Maternal Morbidity and Improved Perinatal Results of Magnesium-Free Tocolysis and Classical Hysterotomy in Fetal Open Surgery for Myelomeningocele Repair: A Single-Center Study

**DOI:** 10.3390/biomedicines11020392

**Published:** 2023-01-28

**Authors:** Mateusz Zamłyński, Anita Olejek, Ewa Horzelska, Tomasz Horzelski, Jacek Zamłyński, Rafał Bablok, Iwona Maruniak-Chudek, Katarzyna Olszak-Wąsik, Agnieszka Pastuszka

**Affiliations:** 1Department of Gynecology, Obstetrics and Oncological Gynecology, Bytom, Medical University of Silesia, 41-902 Katowice, Poland; 2Department of Neonatology and Neonatal Intensive Care, Faculty of Medical Sciences in Katowice, Medical University of Silesia, 40-752 Katowice, Poland; 3Department of Pediatric Surgery and Urology Faculty of Medical Sciences in Katowice, Medical University of Silesia, 40-752 Katowice, Poland

**Keywords:** open fetal surgery, fetal surgery, spina bifida, maternal complications, myelomeningocele

## Abstract

Fetal and maternal risks associated with open fetal surgery (OFS) in the management of meningomyelocele (MMC) are considerable and necessitate improvement. A modified technique of hysterotomy (without a uterine stapler) and magnesium-free tocolysis (with Sevoflurane as the only uterine muscle relaxant) was implemented in our new magnesium-free tocolysis and classical hysterotomy (MgFTCH) protocol. The aim of the study was to assess the introduction of the MgFTCH protocol in reducing maternal and fetal complications. The prospective study cohort (SC) included 64 OFS performed with MgFTCH at the Fetal Surgery Centre Bytom (FSCB) (2015–2020). Fetal and maternal outcomes were compared with the retrospective cohort (RC; *n* = 46), and data from the Zurich Center for Fetal Diagnosis and Therapy (ZCFDT; *n* = 40) and the Children’s Hospital of Philadelphia (CHOP; *n* = 100), all using traditional tocolysis. The analysis included five major perinatal complications (Clavien-Dindo classification, C-Dc) which developed before the end of 34 weeks of gestation (GA, gestational age). None of the newborns was delivered before 30 GA. Only two women presented with grade 3 complications and none with 4th or 5th grade (C-Dc). The incidence of perinatal death (3.3%) was comparable with the RC (4.3%) and CHOP data (6.1%). MgFTCH lowers the risk of major maternal and fetal complications.

## 1. Introduction

Myelomeningocele (MMC) is a non-lethal, severe congenital defect of the neural tube. Its incidence has been estimated at 2–6/10,000 live births [1]. Postnatal MMC repairs are delayed and the outcomes often remain unsatisfactory, as they are associated with intrauterine development of Chiari malformation type II (CMTII), presenting as fetal ventriculomegaly, progressive hindbrain herniation and loss of motor function in the lower extremities, as well as neurogenic bladder and bowel dysfunction [2,3]. Additionally, postnatal intervention requires a timely MMC closure and, in 80% of cases, postnatal placement of the ventriculoperitoneal shunt (VPS), which evacuates excess cerebrospinal fluid (CSF) from the cerebral ventricles into the peritoneal cavity [4]. VPS may be associated with numerous complications, presenting as infections or clogging of the shunting system [5]. Shunt replacement procedures, laminectomy, and decompression of the posterior cranial fossa result in lowered IQ, decreased chances for unassisted living, and verbal and sensorimotor deficiency in the affected patients, as well as the lower overall well-being of the patients and their families [6,7].

At the end of the 1990s and the beginning of 2000, the results of non-randomized clinical trials were published and hinted at significant benefits to prenatal MMC repair, associated with halted progression of CMTII changes [8]. At the same time, significant maternal and fetal risks associated with open fetal surgery (OFS) were emphasized [9]. The prospective, randomized Management of Myelomeningocele Study (MOMS), conducted at three fetal surgery centers in the USA, compared 91 vs. 91 pre- and postnatal OFS MMC repairs and found that prenatal OFS improved neurofunctional outcomes, including the ability to walk at 30 months of age, and lowered the shunting rates [10]. Maternal morbidity and the risk of perinatal complications remain the chief limitations of OFS for MMC repairs. The aim of the study was to evaluate the maternal and fetal outcomes after introducing a modified tocolytic protocol and surgical technique to reduce complication rates.

## 2. Materials and Methods

### 2.1. Study Design

In this study, we compared maternal morbidity and the incidence of OFS risk factors affecting perinatal outcomes in two non-randomized cohorts from the Fetal Surgery Center Bytom (FSC-Bytom; FCSB), with the cohorts from the Zurich Center for Fetal Diagnosis and Therapy (ZCFDT) and the Management of Myelomeningocele Study (MOMS) [11,12,13]. The outcomes of the retrospective cohort (RC) of patients who underwent OFS with a uterine stapler were presented in a study by Zamłyński J et al. [14].

A prospective study cohort (SC) of 64 patients was selected out of 198 candidates presenting at FSCB, or during online consultations, between January 2015 and November 2020. A total of 60 (30%) patients were deemed eligible for prenatal MMC repair; the remaining cases were as follows: 112 (58%)—postnatal management, 9 (4.2%)—termination of pregnancy, 4 (2%)—patient decision to withdraw from the surgery, and 13 (6%)—lost to follow-up after online contact. The patients from the SC underwent OFS with classical hysterotomy protocol, without the use of a uterine stapler, after obtaining written consent (approved by Bioethics Committee of the Medical University of Silesia in Katowice—Statement No. NN-013-296-/I/02/). MMC repairs were performed under general anesthesia and continuous epidural anesthesia, without removing the catheter to facilitate postoperative analgesia. Pfannenstiel incisions were used in patients with a body mass index (BMI) of <30 kg/m^2^. A lower midline incision was used in patients with a history of one—3/58 (5.1%), or two—2/58 (3.4%), cesarean sections (CS). Ultrasounds were used to establish the position of the placenta and the fetus. MMC repairs were performed without exteriorizing the uterus, unless the placenta was located on the anterior wall, in which case the uterus was always exteriorized. If necessary, cephalic version was used to maneuver the fetus into a cephalic position, without applying pressure to the placental site, after profound uterine relaxation was achieved with Sevoflurane, always before the hysterotomy.

A diode laser beam (Leonardo, Biolitec biomedical technology GmbH, Jena, Germany) was used for the initial incision of <1 cm, approximately >5 cm from the placental edge. The amniotic fluid was gradually removed and the amniotic cavity was infused with heated 0.9% NaCl crystalloid solution (400–800 mL). Two parallel DeBakey clamps (B. Braun Melsungen AG, Melsungen Hesse, Germany) were placed, and the uterine muscles and the amniotic membranes were incised to widen the hysterotomy. Before closure, a continuous suture was used to attach the amniotic membranes to the uterine wall and a two-layer continuous suture was used to suture the uterine muscle. The main stages of the classical hysterotomy are presented in Figure 1. A detailed description of the classical hysterotomy protocol has been presented in a recent publication by Zamłyński M et al. [15]. ‘Magnesium-free tocolysis and classical hysterotomy’ (MgFTCH) was selected as the name for the protocol, to include the main modifications. The Clavien-Dindo classification (C-Dc) was used to assess maternal morbidity after MMC repair, on a 5-point scale of perioperative adverse events, classed as mild: grade I—not requiring any pharmacological treatment or surgical intervention (with the exception of analgesic, antipyretic, and antiemetic drugs); and grade II—requiring pharmacological treatment, and severe: grade III—requiring surgical intervention; grade IV—life-threatening complications requiring intensive care (IC) management; and grade V—maternal death [16].

Additionally, we assessed the incidence of the five most common risk factors for MMC-repair-related complications up to ≤34.0 GA: preterm labor (PTL), premature rupture of membranes (PROM), chorion-amniotic separation (CAS), placental abruption, and hysterotomy site assessed during CS (NIH; non-intact hysterectomy). The results in the prospective cohort were compared with our retrospective OFS findings, published elsewhere, where anesthesia, tocolysis, and hysterotomy were performed using a uterine stapler, in accordance with the MOMS protocol [13]. The prospective and retrospective study cohorts included patients deemed eligible using the following MOMS criteria: maternal age >18 years, singleton pregnancy between 19.0 and 25.6 GA, no abnormal karyotype or, other than MMC, anatomical abnormalities, and MMC located at S_1_ or above, with a hernia sack.

The exclusion criteria due to maternal reasons were as follows: cervical insufficiency and/or short cervix (<20 mm on ultrasound), history of PTL (delivery at <37 GA), inherited Mullerian duct anomalies, postoperative uterine deformations, type 1 diabetes, maternal-fetal Rho(D) isoimmunization, obesity defined as a BMI of ≥35 kg/m^2^, anti-HIV or anti-HCV seropositivity, HBV viremia, other serious maternal illnesses, psychosocial problems, lack of patient support, and/or inability to travel and participate in the follow-up. Outside of the MOMS criteria, four women with pregnancies uncomplicated by obesity (range: BMI 35.0–42.1 kg/m^2^), of CMTII and acceptable kyphosis (>30 degrees) were deemed eligible for MMC repair [10].

Fetal karyotype was determined from the amniotic fluid cells using microarray-based comparative genomic hybridization (aCGH). Standardized ultrasound and magnetic resonance imaging (MRI) were used to establish the location of the upper part of the defect and the fMMC structure, the atrial diameter and the hindbrain position. All patients started the therapy with nifedipine (Adalat, Bayer Vital GMBH, Leverkusen, Germany), a calcium channel blocker, at a daily oral dose of 30 mg, one week before fMMC repair, and continued at a dose of 60 mg until delivery. Therapy with COX-1/2 inhibitor was initiated one day before the repair and continued for three days at an oral dose of 75 mg, twice daily (Metindol retard, PharmaSwiss/Valeant, Prague, Czech Republic). No routine preoperative intravenous tocolysis was used; only an anesthesiologic agent, a fluorinated methyl isopropyl ether known as Sevoflurane (1,1,1,3,3,3-Hexafluoro-2-propanol) (1–3%) (Sevoflurane Baxter, Warsaw, Poland) was used during MMC repair to lower the tension of the uterine smooth muscles. Occasionally, magnesium sulphate, Atosiban, or fenoterol hydrobromide were used intravenously because of uterine spasms or hypertonicity during the procedure. No routine postoperative intravenous tocolysis was used. In the event of uterine contractility, Atosiban was administered as a 6.75 mg bolus intravenously (IV), followed by 18 milligrams per hour (mg/h) over 3 hrs and 6 mg/h for the following 45 h (Tractocile, infusion of 37.5 mg/5 mL) (Ferring AG, Baar, Switzerland)—this was used as first-line tocolysis for up to 2 cycles, or a β-2 mimetic (Teva Pharmaceuticals, Warsaw, Poland), 1–3 µg/min/48 h, was used in the second cycle. The effectiveness of the tocolysis and fetal well-being were monitored using continuous CTG (Monako ITAIM, Zabrze, Poland).

Intraoperatively a crystalloid solution was used for hydration, the volume depended on the following: BMI values, loss of body fluids, duration of the MMC repair, and other factors, as described in the anesthesia protocol and standards for fetal surgery protocols [17,18].

Postoperatively, all women were placed in IC recovery rooms for 48 h to monitor maternal parameters and fetal well-being (USG, CTG). Enoxaparin sodium (40 mg/24 h/s.c.) (Sanofi Aventis, Warszawa, Poland) and compression therapy were used as antithrombotic measures from preoperative day 1, to complete patient mobilization on postoperative day 4. The therapy was continued until delivery in patients with a BMI of ≥35 kg/m^2^.

### 2.2. Statistics

The analyses were performed using the MedCalc software Ver 20.110 (MedCalc Software Ltd., Ostend, Belgium). Data are expressed as percentages or means with standard deviations (SD), or medians with quartiles. The χ2 and χ2 for tests for trends were used to compare variables between the investigated groups. In addition, the odds ratios (OR) with 95% confidence intervals (CI) were calculated. A *p*-value of <0.05 was considered statistically significant.

## 3. Results

A comparative analysis of patient characteristics in the SC and three reference groups deemed eligible for MMC repair is presented in Table 1. The main clinical data are comparable in all cohorts.

In the SC, four patients had a BMI of 38–41 kg/m^2^; mean GA at delivery was 36.1 ± 2.0 (30 6/7–38.3 6/7) wks; there were no cases of delivery <30.0 GA; 36/58 (61%) patients delivered >34.6 GA. Mean operative time for MMC repair (skin to skin) was 131 ± 32 min. The mortality rate was 2 (3%), including 1 intrauterine and 1 neonatal death within 24 h after the MMC repair.

Complications according to the Clavien-Dindo classification are presented in Table 2.

Grade 1 complications were found in 11 (18.3%) patients with at least 1 or 2 mild complications. Grade 2 complications resulting from MMC repair were statistically significantly less common in the SC than in the ZCFDT and CHOP cohorts—oligohydramnios was found in 7/58 (12%) in the SC, compared to 12 (30%, *p* < 0.001) and 22/96 (22.9%, *p* < 0.005) of the ZCFDT and CHOP cases, respectively. All cases of CAS, 4/22 (18%), developed at 30 + 0 − 34 + 6 GA, three of them after 2–3 days, which led to massive PROM, 4/22 (36.3%), and the need to end the pregnancy >30.0 GA.

Grade 2 complications were the consequence of using intravenous tocolysis (β-2 mimetic) for uterine contractility, intra- or postoperatively. The use of tocolytic agents was statistically significantly less common in the SC compared to the RC (1 (1.7%) vs. 40 (87%), respectively; *p* < 0.001). In the SC, PROM was statistically significantly less common than in the RC (11/58 (19%) vs. 4 (52.2%); *p* < 0.01). In addition, spontaneous contractions of the uterine muscle at <37 GA were statistically significantly less common in the SC (10/58 (17.2%)) than in the RC (26 (56.5%); *p* < 0.001), and in the CHOP cohort (36/96 (37.5%); *p* < 0.01). Intraoperative use of magnesium sulfate was necessary in only four patients (6.6%), which is statistically significantly lower compared to the uterine stapler method used with the remaining three cohorts: RC, 23 (50%), *p* < 0.001; ZCFDT, 15 (37.5%), *p* < 0.001; and CHOP, 100 (100%), *p* < 0.001. Magnesium sulfate, with a 6 g loading dose (bolus), was administered before the initial incision of the uterus due to excessive uterine muscle tension. In the SC, tocolytic treatment due to uterine contractility at <37 GA was implemented using intravenous infusion of Atosiban in 10 (16.6%) women with PROM, to gain enough time for repeat corticosteroid therapy, according to the antenatal steroid prophylaxis protocol.

As for severe grade 3 complications, three cases were observed: one (1.7%) multipara, with a BMI of 35 kg/m2, requiring revision of the abdominal scar due to a progressive subfascial hematoma at 31 GA; one (1.7%) case of severe extragenital bleeding, leading to placental abruption; one (1.7%) case of CAS, which eventually led to placental abruption; and one (1.7%) case of the placenta located on the anterior wall, necessitating an emergency CS at 33.4 GA.

No complications requiring IC/ICU management (grade 4) or cases of maternal mortality (grade 5) were observed.

Maternal morbidity, according to the Clavien-Dindo classification, in combined mild (grades 1–2) and severe (grades 3–5) complications in the SC, was compared with the global reports for open surgery according to the MOMS protocol and fetoscopic MMC repair published by Sacco et al. [19] and presented in Table 3. Statistical analysis revealed no statistically significant differences in maternal morbidity in the SC as compared to the main groups and types of complications.

A comparison of six main complications which developed before 34 GA in the SC, RC, and CHOP cohorts is presented in Table 4. The incidence of CAS in the SC was lower than in the CHOP cohort (OR = 0.28; 95% CI: 0.08–0.95). The incidence of PROM in the SC was lower than in the RC and CHOP cohorts: OR = 0.19 (95% CI: 0.06–0.62) and OR = 0.24 (95% CI: 0.07–0.75), respectively. Similarly, spontaneous contractions of the uterine muscle were less frequently observed in the SC than in the RC and CHOP cohorts: OR = 0.29 (95% CI: 0.10–0.91) and OR = 0.09 (95% CI: 0.03–0.29), respectively.

## 4. Discussion

To the best of our knowledge, approximately 1281 cases of OFS were reported and 373 fetoscopic MMC repairs were conducted between 1997 and 2018. OFS are performed in 23 (67.6%) centers and fetoscopic surgeries in five (14.7%) centers, while six (17.6%) centers perform both types of MMC repairs [20]. Numerous publications on the results of surgical techniques used in cohort groups describe two main techniques of prenatal MMC repair: OFS and fetoscopy. In a new systematic review, Giradelli et al. indicated that only 20 publications had compared data on maternal morbidity and short-term fetal assessment in accordance with the MOMS inclusion and exclusion criteria [21]. The comparison of the results of these studies is limited by diverse surgical techniques and a lack of randomization.

Risk assessment of maternal morbidity remains an important aspect of these procedures. Therefore, in this study we evaluated the incidence of maternal and perinatal complications, from surgery to delivery, using the Clavien-Dindo scale. We used the MgFTCH protocol, with modified tocolytic treatment and hysterotomy, rather than the standard MOMS protocol [10].

### 4.1. Modifications to the Tocolytic Treatment

In our modified tocolytic treatment protocol, we decided to avoid routine administration of magnesium sulphate. The use of large intravenous doses of magnesium sulphate in the MOMS protocol has been described by Ferschl et al. [17,22]. A total perioperative dose of 6 g, followed by a postoperative dose of 24 g during a 24 h period, is borderline toxic [11]. In the first study by Golombeck et al., a retrospective review of 87 hysterotomies performed for fetal surgery in a single institution between 1989 and 2003, 28% of the mothers developed pulmonary edema [6]. Adzick et al., in a randomized study, reported a 6% incidence (5/78) of pulmonary edema. Earlier observations, from our center and those of other authors, estimated the incidence of that complication, which depends on intravenous doses of magnesium sulphate, at 2.2–5.5% [10,13]. Two meta-analyses of perioperative tocolytic lines and anesthesiologic procedures, found differences in complications after using magnesium sulphate, β-mimetics, indomethacin and nifedipine, which is a limitation when attempting to determine the cause of complications in maternal-fetal surgery [23,24]. Ochsenbein-Kölble et al. replaced magnesium sulphate in the first line of tocolysis with Atosiban (an inhibitor of oxytocin and vasopressin) in equal-sized groups (n = 15, each) [25]. According to these authors, 6% of the patients developed complications related to intravenous administration of magnesium sulphate, at serum levels of 3.2 ± 0.4 mmol/L, associated with electrolyte imbalance manifesting as decreased (33–40%) levels of calcium, sodium, and potassium, including 100% neurological symptoms and right bundle branch block. In contrast, the use of Atosiban was not associated with the development of neurological symptoms and pulmonary edema. Perinatal outcomes were comparable in both groups [25]. In our study, intravenous infusion of Atosiban was postoperatively administered in 10.3% (6/58) of the cases, due to hypertonicity or contractile activity. As in the ZCFDT cohort, we also observed no dependent complications, as classified using the C-Dc. Our MgFTCH protocol did not include routine pre- or intraoperative intravenous tocolysis with magnesium sulphate. Magnesium sulphate for tocolysis was administered perioperatively in only 6.7% of the patients, which was statistically significantly less frequent than in the RC, ZCFDT and CHOP cohorts (50%, 37.5%, and 100%, respectively). In our study, the need to use a tocolytic agent was associated with perioperative placental ablation (1 case) and hypertonicity without uterine contractility (3 cases). Profound intraoperative relaxation of the uterine muscle was achieved with an inhalational anesthetic, Sevoflurane (1–3%) (Sevoflurane, Baxter). In our protocol, we significantly decreased the use of intravenous tocolysis, which was administered to only 10.3% of the surgical pregnant patients during the steroid therapy between 29.4 and 36.0 wks GA. In our study, the modifications to the tocolytic protocol did not result in the development of tocolysis-dependent complications (grades 1–5), as described in the C-Dc.

In addition, the MgFTCH protocol includes indomethacin, a non-selective inhibitor of prostaglandin cyclooxygenase (COX-1/2), because it inhibits the conversion of free arachidonic acid to PGE_2_ and PGF_2α_ [26]. Contractile activity of the myometrium is induced by PGE_2α_ using the following pathway: expression of the oxytocin receptor, increased formation of connexin-43 gap junctions, and enhanced transport of free Ca^2+^ from the endoplasmic reticulum [27,28]. The total indomethacin dose administered to our SC was the highest (450 mg/3 days), compared to the MOMS cohort (300 mg/2 days), the CHOP cohort (250 mg/3 days), and the Fetal Center at Vanderbilt (50 mg/1 day) [10,13,29]. Only one patient (1.7%) presented with symptoms of gastritis (grade 1), and symptom resolution was achieved after 1 day of therapy with an intravenous proton-pump inhibitor. Indomethacin crosses the maternal-fetal barrier at a low gradient of concentration (4:1) [30]. The use of COX-1/2 inhibitors may result in suppression of urine output and is associated with a risk of closure of the fetal ductus arteriosus. COX-1/2 synthase inhibitor impedes the production of fetal prostacyclin, which is necessary to maintain low pressure of the fetal renal arteries, and consequently, urine production. Inhibition of prostacyclin production, induced by indomethacin, results in increased renal arterial resistance and decreased urine production, which in normal conditions is the source of high prostacyclin concentrations in the amniotic fluid [31]. Up until three weeks after OFS, we diagnosed oligohydramnios which was not associated with amniotic fluid leakage or PROM in seven (12%) cases.

We demonstrated that the incidence of oligohydramnios in the SC was comparable to that of the other cohort groups. We also observed a comparable incidence of oligohydramnios and delivery at <34 GA in the cohort groups: SC at 31.8% vs. RC at 11.1%, and SC at 31.8% vs. CHOP at 42.2%. In our study, we demonstrated that the duration of oligohydramnios did not exceed 21 days after the last dose of indomethacin, which is consistent with our earlier reports [14,15]. Our observations during the neonatal period were not indicative of indomethacin nephrotoxicity.

Additionally, in accordance with the MOMS protocol, the patients from our study and retrospective cohorts received oral doses of nifedipine, a calcium channel blocker, at a daily dose of 30–80 mg/24 h, before MMC repair up until the delivery [10]. Ochsenbein-Kölble et al. reported using high doses of the medicine (up to 120 mg/24 h from postoperative day 1 to delivery), which were well-tolerated by the affected patients [24]. In contrast, Sacco et al. used a calcium channel blocker after surgery up until the end of the hospitalization period [32]. We confirm that nifedipine may be used as a third-line tocolytic agent, supporting intravenous tocolysis during hospitalization. Additional benefits of nifedipine include cardioprotective activity, stabilization of blood pressure, inhibition of blood platelet aggregation, and the possibility of using it in ambulatory care [33]. According to our observations, long-term use of nifedipine at a daily dose of up to 80 mg/24 h is not associated with the development of complications described in the C-Dc.

### 4.2. Modifications in Hysterotomy

The current surgical technique for hysterotomy, with the use of a surgical stapler and a two-layer closure of the uterine muscle, has been described in a randomized MOMS study [10]. Staple line dehiscence, with the subsequent blood loss and chorioamniotic membrane separation requiring additional repair at the hysterotomy site, is a limitation of that technique. The healing process of the uterine scar consists of connecting the surgical staples in a humid environment, and creating a space that is resistant to uterine muscle stretching in the subsequent course of pregnancy. In our 2014 publication, we demonstrated the obtained perinatal and maternal results. Despite significant acquisition of experience of the surgical team, our 2005–2015 observations indicated that the learning curve presents a flat value [34]. In our protocol, we used the classic hysterotomy procedure during a three-layer closure of the uterine muscle, as described by Moron et al. [35]. Our previous observations indicated that classic hysterotomy allows for complete visual control of the uterine opening. Double fixation of the amniotic membrane to the uterine muscle prevents the development of CAS and effectively isolates the chorion from the leaking amniotic fluid, which contains high concentrations of prostaglandins (Figure 1) [15,35,36]. The healing of the hysterotomy scar occurs in the anatomical layers of the uterine muscle. Carvalho et al. performed a histological evaluation of ten chorioamniotic membranes, obtained after CS, from pregnancies which underwent OFS for MMC repair [37]. The picrosirius red sign was more intense at the suture site, which was associated with the localization of collagen type 1. The number of collagen fibrils at the suture site was statistically significantly higher than at the non-suture site (13.22 ± 2.84 vs. 6.16 ± 1.09, respectively; *p* < 0.0001) [37]. Enhanced collagen synthesis with tissue restoration is suggestive of reparative activity at the site of the closing suture for the chorioamniotic membrane, a mechanism which most likely prevents amniotic fluid leakage, allowing continuation of the pregnancy after OFS [37]. Johnson et al. determined the incidence of complications in 45 out of 91 pregnant women who underwent hysterotomy during OFS with a uterine stapler and delivered at, or before, 34 weeks of gestation [13]. In their analysis of several parameters, these authors found no statistically significant risk factors for non-intact hysterotomy (NIH). Our findings indicate that NIH incidence in the SC patients undergoing classical hysterotomy was statistically significantly lower than in the randomized CHOP data (13.6% vs. 68.8%, respectively; OR = 0.20; 95% CI: 0.07–0.58). We hypothesize that classical uterine surgery in prenatal MMC repair promotes timely healing of the post-hysterotomy scar.

The main goal of the modified surgical technique was to prevent or lower the incidence of CAS, which is a severe complication, and the related consequences, i.e., PROM and PTL. In 2016, Botelho et al. described a surgical technique of mini-hysterotomy, the main goal of which was to reduce traumatization of the uterine muscle by shortening the incision line to 3.05 cm and sewing amniotic membranes with a continuous suture into the open muscle, in order to prevent CAS [38]. As a result, one patient (1/39; 2.6%) experienced CAS. Nine patients (9/39; 23.1%) had preterm rupture of membranes at a median GA of 34.1 weeks (range: 31.1–36.0). The average GA at delivery was 35.3 weeks (SD: 2.2; range: 27.9–39.1). Ninety-five percent (37/39) of our patients had an intact hysterotomy site at delivery. Additionally, in a retrospective cohort of 109 patients who underwent OFS with the early mini-hysterotomy technique between 19.7 and 26.9 GA, Peralta et al. showed a reduction in the need for VPS (*p* = 0.049), higher rates of hindbrain herniation reversal (*p* = 0.003), and longer latencies from surgery to delivery (*p* < 0.001). Median GA at delivery was 35.3 weeks [39]. Soni et al. reported the incidence of CAS and PPROM (preterm premature rupture of membranes) after MMC repair to be 23.9% and 30.7%, respectively [40]. In that study, local or global CAS, as diagnosed by ultrasound, was statistically significantly more often associated with PPROM than in pregnant women with intact chorioamniotic membranes (59.1% vs. 21.2%). Additionally, CAS corresponded with the need to deliver the newborn statistically significantly earlier than in the group without CAS (32.1 ± 4.2 vs. 34.4 ± 3.5 weeks, respectively). The values did not depend on whether the placenta was located on the uterine wall. In contrast, Corroenne et al., in their retrospective cohort study of 91 patients after fetoscopic (laparotomy and exteriorized uterus) or open hysterotomy MMC repair, identified placenta located on the uterine wall as an independent risk factor for postoperative development of CAS, OR = 3.72; 95% CI: 1.46–9.5 [41]. The localization of the placenta on the anterior uterine wall significantly increased the risk for CAS after fetoscopic repair, OR = 3.94; 95% CI: 1.14–13.6. CAS diagnosed at >30 wks GA was a risk factor for preterm labor, compared to CAS diagnosed at ≥30 wks GA (90% vs. 36%, respectively; *p* = 0.01). In that study, the area of the uterus affected by CAS ranged from 25% to >50%. In our study, the size of the separation did not exceed 5 cm and was located between two parallel lines of continuous sutures (Figure 1). Additionally, in all cases CAS developed late in pregnancy, at >30.0 wks GA. None of the pregnancies needed to be delivered before 30 wks GA.

## 5. Conclusions

The outcomes of our MgFTCH protocol indicate that perioperative routine administration of intravenous magnesium sulphate for tocolysis is not necessary. Sevoflurane is a sufficient muscle relaxant during an MMC repair.

The classical hysterotomy technique used in the MgFTCH protocol focused on achieving watertight closure of the amniotic membranes, in order to prevent amniotic fluid leakage, development of CAS, PROM and the resulting PTL. The lack of cases of perinatal complications resulting from NIH, such as CAS, PROM and PTL, up to 30 wks GA, is an especially beneficial outcome of implementing the MgFTCH protocol. Key points for implementation of the MgFTCH protocol were stated in Table 5.

## Figures and Tables

**Figure 1 biomedicines-11-00392-f001:**
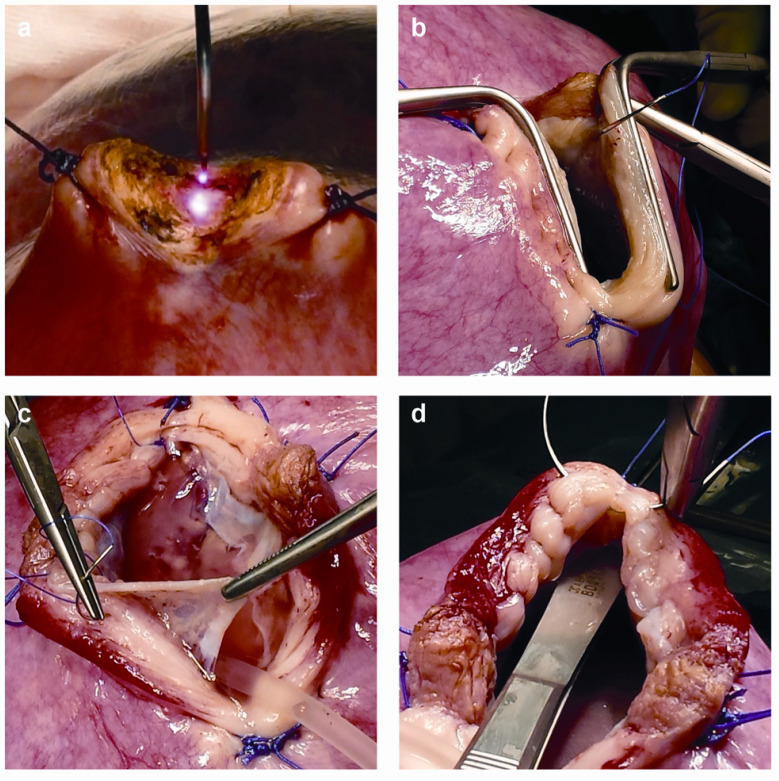
Technical details of intrauterine MMC repair (IUMR): (**a**) Initial opening of the uterus with a diode laser. (**b**) Placement of the DeBakey clamps in the line of the incision and fixation of the amniotic membrane to the uterus with continuous suture. The total length of the uterine opening in complete relaxation is 66 mm (initial incision of 10 mm and final—working length between DeBakey clamps is 56 mm). (**c**) Closure of the amniotic cavity with amniotic membrane fixation to the incised edge of the uterus with continuous suture. The length of the uterine opening is only 45 mm due to increased uterine tone. (**d**) The first layer of continuous intramural sutures of the uterus.

**Table 1 biomedicines-11-00392-t001:** Comparison of patient demographic and clinical characteristics from all cohorts.

	Study Cohort FSCB *n* = 60	Retrospective Cohort FSCB *n* = 46	Post-MOMS (CHOP) *n* = 100	ZCFDT *n* = 40
Parity, *n* (%)				
Nulliparous	35 (59)	21 (45.6)	35 (35) **	21 (52.2)
Maternal age at screening, years, mean ± SD (range)	30 ± 5 (19–40)	29 ± 5	29.7 (18–41)	29.9 ± 4.9
GA at evaluation, (wks/d), mean (range)	23.4 ± 1.1 21 1/7–26 6/7	20–25	21 6/7 18 1/7–25 4/7	NA
Ethnicity, *n* (%)				
Caucasian	60 (100)	46 (100)	88 (88) **	37 (92.5) *
Other	0	0	12 (12)	3 (7.5)
BMI (kg/m^2^), mean ± SD (range)	27.6 ± 4.8 (16–40)	24.1 ± 3.2	26.3 (18.7–35)	27.2 ± 25.4
Smoking	5 (8.3)	4 (8.7)	NA	0
Fetal sex, *n* (%)				
Female	28 (46.6)	23 (50)	52 (52)	23 (53)
GA at fetal surgery (wks), mean ± SD	24.2 ± 7.3	24.7 ± 7.9	23.3 (20 2/7–25 6/7)	24.7 ± 0.84
Total operative time (skin to skin) (min), mean ± SD (range)	131 ± 32	NA	78 (54–106)	139 ± 24
Gestational age at birth (wks), mean ± SD (range)	36.1 ± 2.0 (30 6/7–38.3 6/7)	NA	34.3 (22 1/7–37 4/7)	35.5 ± 2.3
<30 GA, *n* (%)	0	15/44 (34.1)	9/96 (9.4)	1 (2.5)
30.0 to 34.0 GA, *n* (%)	22/58 (38)	11/44 (25)	35/96 (36.4)	10 (25)
34.1 to 36.6 GA, *n* (%)	18/58 (31)	10/44 (22.7)	26/96 (27.1)	15 (37.5)
≥37 GA, *n* (%)	18/58 (31)	8/44 (18.2) ^^^	26/96 (27.1)	14 (35)
Fetal birth weight (g), mean ± SD (range)	2.620 ± 511 (780–3.820)	NA	2.415 (501–3.636)	2.631 ± 535
Perinatal death, *n* (%)	2 (3.3) 1 IUFD 1 NND	2 (4.3) 1 IUFD 1 NND	6/98 (6.1) 2 IUFD 4 NND	NA

Statistical significance vs. Study Cohort (* *p* < 0.05; ** *p* < 0.01). ^^^ include all categories of the variable. Abbreviations: GA—gestational age; wks—weeks; d—days; BMI—body mass index; g—grams; min—minutes; IUFD—intrauterine fetal demise; NND—neonatal death

**Table 2 biomedicines-11-00392-t002:** Comparison of operative and perinatal complications according to the Clavien-Dindo classification in all cohorts.

	Study Cohort FSCB *n* = 58/60	Retrospective Cohort FSCB *n* = 44/46	ZCFDT *n* = 40	Post-MOMS (CHOP) *n* = 100/98
*Grade 1—mild complications not requiring any pharmacological treatment or surgical intervention*				
Gestational diabetes (nutritional therapy)	2 (3.3)	1 (2.2)	6 (15) *	NA
Seroma	4 (6.6)	NA	10 (25) *	
Hematoma	0	1 (2.2)	3 (7.5) *	
Wound dehiscence (skin)	2 (3.3)	1 (2.2)	2 (5)	
Symptoms of subileus (nausea/flatulence)	1 (1.7)	0	0	
Symptomatic cholecystolithiasis (analgesics only)	0	0	1 (2.5)	
Transient amniotic fluid leakage	1 (1.7)	3 (6.5)	4 (10)	
CAS	4 (6.8)	8 (17.3)	12 (30) **	22/96 (22.9) *
*Grade 2—complications requiring pharmacological treatment*				
Gestational diabetes type 1	0	0	2 (5)	NA
Pulmonary edema (no intubation)	0	1 (2.1)	0	2 (2)
Pregnancy-induced hypertension	1 (1.7)	0	0	0
Urinary tract infection	1 (1.7)	1 (2.1)	0	NA
Cholestasis of pregnancy	0	0	1 (2.5)	
Thrombotic disease	1 (1.7)	0	0	
Blood transfusion	1 (1.7)	3 (6.5)	0	8 (8.8)
Oligohydramnios (AFI < 5 cm)	7/58 (12)	4 (8.7)	NA	6/96 (6.3)
Tocolytic treatment for uterine contractility (intra or post fMMC repair)				
Beta_2_ mimetic	1 (1.7)	40 (87) ***	0	NA
Atosiban	10 (16.6)	NA	25 (62.5) ***	NA
Magnesium sulfate	4 (6.6)	23 (50) ***	15 (37.5) ***	100 ***
Hexoprenaline	0	0	NA	0
Nifedipine p.o.	60 (100)	38 (82.6) **	40 (100)	100
COX-1/2 inhibitors	60 (100)	40 (87) **	40 (100)	100
PROM	11/58 (19)	24 (52.2) **	14 (35)	31/96 (32.3)
Spontaneous contractions of the uterine muscle <37^+0^ weeks	10/58 (17.2)	26 (56.5) ***	NA	36/96 (37.5) **
*Grade 3—complications requiring surgical intervention*				
Peritonitis	0	1 (2.2)	0	0
Seroma (requiring surgical intervention)	0	0	2 (5)	NA
Hematoma (requiring surgical intervention)	1 (1.7)	0	0	
Cholecystolithiasis (requiring surgical intervention)	0	0	1 (2.5)	0
Pre-eclampsia/Eclampsia	0	2 (4.3)	1 (2.5)	1 (1)
Chorioamnionitis (not requiring IC/ICU management)	0	2 (4.3)	1 (2.5)	4 (4)
Incisional hernia	0	0	1 (2.5)	0
Bartholin’s cyst	0	0	1 (2.5)	0
Placental abruption	1 (1.7)	2 (4.3)	4 (10)	6 (6.6)
Major bleeding (extragenital)	1 (1.7)	2 (4.3)	1 (2.5)	0
Uterine rupture/Fetal extrusion into the peritoneal cavity	0	1 (2.2)	0	0
*Grade 4—life-threatening complications requiring IC/ICU management*				
Third-degree AV block with mechanical reanimation/other acute heart disease	0	0	1 (2.5)	0
Lung embolism	0	0	1 (2.5)	0
Urosepsis	0	0	0	0
Pulmonary edema (with intubation)	0	0	1 (2.5)	0
Uterine rupture	0	0	1 (2.5)	0
Chorioamnionitis	0	0	1 (2.5)	0
*Grade 5—maternal death*	0	0	0	0

Statistical significance vs. Study Cohort (* *p* < 0.05; ** *p* < 0.01; *** *p* < 0.001). Abbreviations: CAS—Chorioamniotic membrane separation; AFI—amniotic fluid index; PROM—premature rupture of membranes; IC—intensive care; ICU—intensive care unit; AV—atrio-ventricular.

**Table 3 biomedicines-11-00392-t003:** Maternal morbidity in the study group compared to the global data of the fetoscopic technique or open surgery of MMC, according to the Clavien-Dindo classification.

Clavien-Dindo Classification	Severe Complications				Mild Complications		
	IV—Requiring ICU Care		III—Requiring Surgical Intervention		I—Requiring Only Analgesic, Antipyretic, and Antiemetic Drugs II—Requiring Pharmacological Treatment		
**Open MMC repair** ***n* = 779**	Severe infection	2 (0.2)	Hemorrhage requiring delivery	3 (0.4)	Bleeding during procedure	1 (0.1)	**ALL COMPLICATIONS: 11.54%** **(95% CI, 7.73–15.99)**
	Complete heart block	1 (0.1)	Placental abruption	16 (2.0)	Transfusion during/after procedure	5 (0.6)	
	Pulmonary edema	1 (0.1)	Bowel obstruction	1 (0.1)	Chorioamnionitis	21 (2.7)	
			Uterine rupture	4 (0.5)	Other infections	2 (0.2)	
			Cesarean hysterectomy	1 (0.1)	Pulmonary edema	15 (1.9)	
					Transfusion at delivery	16 (2.0)	
	Total grade IV n, (%)	4 (0.5)	Total grade III *n*, (%)	25 (3.2)	Total grade I–II *n*, (%)	60 (7.7)	
	**TOTAL SEVERE: 3.35%** **(95% CI, 1.70–5.53)**				**TOTAL MILD: 6.63%** **(95% CI, 3.63–10.45)**		
**Cohort study group** ***n* = 58**	Severe infection	0	Hemorrhage requiring delivery	1 (1.7)	Bleeding during procedure	1 (1.7)	**ALL COMPLICATIONS: 10.34%** (**95% CI, 3.80–22.52)**
	Complete heart block	0	Placental abruption	1 (1.7)	Transfusion during/after procedure	0	
	Pulmonary edema	0	Bowel obstruction	0	Chorioamnionitis	0	
			Uterine rupture	0	Other infections	1 (1.7)	
			Cesarean hysterectomy	0	Pulmonary edema	0	
					Transfusion at delivery	2 (3.4)	
	Total grade IV *n*, (%)	0	Total grade III *n*, (%)	2 (3.4)	Total grade I–II *n*, (%)	4 (6.6)	
	**TOTAL SEVERE: 3.45%** **(95% CI, 2.42–7.22)**				**TOTAL MILD: 6.90%** **(95% CI, 1.88–17.66)**		
**Fetoscopic MMC** **repair *n* = 268**	Severe infection	0	Hemorrhage requiring delivery	0	Bleeding during procedure	3 (1.1)	**ALL COMPLICATIONS: 12.49%** **(95% CI, 4.83–23.06)**
	Complete heart block	0	Placental abruption	6 (2.75)	Transfusion during/after procedure	0	
	Pulmonary edema	0	Bowel obstruction	0	Chorioamnionitis	10 (11.2)	
			Uterine rupture	0	Other infections	0	
			Cesarean hysterectomy	0	Pulmonary edema	5 (1.8)	
					Transfusion at delivery	0	
	Total grade IV *n*, (%)	0	Total grade III *n*, (%)	6 (2.75)	Total grade I–II *n*, (%)	18(9.04)	
	**TOTAL SEVERE: 2.75%** **(95% CI, 0.56–6.52)**				**TOTAL MILD: 9.04%** **(95% CI, 3.27–17.40)**		

**Table 4 biomedicines-11-00392-t004:** Comparison of prenatal complications developing up to ≤34.0 GA.

GA at Birth	Study Cohort FSCB		Retrospective Cohort FSCB			CHOP Cohort	Comparison of Prenatal Complications Developing Up to 34 GA	
	30^+^–34^+0^ *n* = 22	<34^+0^ *n* = 22	<30 *n* = 15	30^+^–34^+0^ *n* = 11	<34^+0^ *n* = 36	<34^+0^ *n* = 45	SC vs. RC	SC vs. CHOP cohort
CAS *	0 ^+4 (18.2)	4 (18.2)	7 (46.6)	1 (9.0)	8 (22.2)	20 (44.4)	0.78 (0.20–2.97) ns	0.28 (0.08–0.95) 0.04
Oligohydramnios AFI<5 cm or MVP<1 cm	4 ^+3 (31.8)	7 (31.8)	2 (13.3)	2 (18.1)	4 (11.1)	19 (42.2)	3.73 (0.95–14.43) 0.06	0.64 (0.22–1.87) ns
PROM	1 ^+4 (22.7)	5 (22.7)	12 (80.0)	10 (90.9)	22 (61.1)	25 (55.5)	0.19 (0.06–0.62) <0.01	0.24 (0.07–0.75) 0.01
Spontaneous contractions of the uterine muscle <34 wks	4 ^+3 (31.8)	7 (31.8)	12 (80)	10 (90.9)	22 (61.1)	38 (84.4)	0.29 (0.10–0.91) 0.04	0.09 (0.03–0.29) <0.001
Placental abruption	0 ^+1 (4.5)	1 (4.5)	1 (6.6)	0	1 (2.7)	6 (13.3)	1.67 (0.10–28.08) ns	0.31 (0.03–2.74) ns
Non-intact hysterotomy (NIH)	0 ^+3 (13.6)	3 (13.6)	1 (6.6)	1 (9.0)	2 (5.5)	31 (68.8)	2.68 (0.41–17.51) ns	0.07 (0.02–0.28) <0.001

* CMS—sonographically detected area of the chorioamniotic space, max. size: 4–6 cm. Larger CMS were not observed. Abbreviations: GA—gestational age; CAS—chorioamniotic separation; AFI—amniotic fluid index; MVP, maximal vertical pocket; PROM—premature rupture of membranes; wks—weeks; NIH—non-intact hysterotomy. ^ complications diagnosed before 30.0 wks GA.

**Table 5 biomedicines-11-00392-t005:** MgFTCH protocol–key points for implementation.

Stages of Surgical Hysterotomy	Uterine Muscle Relaxants
Initial incision and placement of the DeBakey clamps with the use of a diode laser.	Perioperative use of nifedipine and indomethacin.
Two lines of opposing continuous sutures along the uterine incision, with homeostasis and CAS prevention.	Anesthesiologic agent, Sevoflurane, is used throughout the entire course of the hysterotomy. The use of magnesium sulphate is not necessary.
Fixation of the amniotic membranes to the site of the uterine incision using continuous suture.	Complete exchange of the amniotic fluid with heated crystalloid solution after the initial incision, with perfusion during the entire course of surgery.
Placement of two continuous sutures ”layer to layer” on the uterine muscle.	Postoperative administration of nifedipine. Atosiban or β-mimetic are used in case of uterine contractility.

## Data Availability

The data presented in this study are available in corresponding references.

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
