# Peer review of "Decreased Maternal Morbidity and Improved Perinatal Results of Magnesium-Free Tocolysis and Classical Hysterotomy in Fetal Open Surgery for Myelomeningocele Repair: A Single-Center Study"

_biomedicines, 2023, doi:10.3390/biomedicines11020392_

Round 1
Reviewer 1 Report
The research study comes from a reliable and established group preforming fetal surgery. The article is overall well-written, interesting and sound. I have some constructive criticisms for the authors to improve the article before publication.
Specific comments
1. There is a lack of discussion of different alternative tecniques. They may cite a recent review comparing all available techniques (ref 1). Each techniques offers benefits and presents problems and most importantly should be adapted to the specific patient undergoing surgery. This information would be interesting for readers in the background and/or discussion. (ref 1)
2. The authors should aknowledge that their technique is neither novel nor innovative and it was already described years before by the brazilian group of CFA Peralta (ref 2-3). The mini histyerotomy by Peralta does not involte use of stapler, presents lower invasiveness on the uterus and an overall reduced rate of prematurity with all other outcomes showing non-inferiority to that of the MOMS trial (ref 2). More recently, the same group showed that operating at earlier gestational ages the risk of postnatal ventriculoperitoneal shunting was reduced (ref 3). This major evidence should be cited and briefly referenced.
3. A clear aim of the study is missing at the end of the introduction both in the abstract and main text. Please add (the aim becomes evident reading the paper but shiuld be stated early in the manuscript). The object is assessing a modified technique that they hypotesize may lead to a reduction of perinatal complications.
4. Please make clear from the title that we are dealing with perinatal complications (neurological outcomes are not shown herein).
5. They should describe more why the surgical technique was changed overtime. If there was a learing curve it is clear that outcome would be better in the more recent cases as compared to the others. In other words, if one technique involve an improvement and the surgeon is more skilled and expert, the outcome would be better. Were these consecutive cases? This should be discussed in the limitation section.
6. It is mot cleaer what do they mean by low-risk fetal surgery. Is the procedure deemed at lower risk of complication? The patients do not look at low-risk of complications. The title as it is would not be scientifically sound since they perform a study to demontrate difference which are not known a priori In the first case they should modify the title removing "low risk", in the second case they should redifine that they selected patients with lower risk. mThe same is true for the table of key points. The key conmcept is that each procedure poresents benefits and risks. There is no sufficient evidence to state that their procedure reduces the risks as compared to the others. Please smooth their conclusions.
Minor
7. Please make clear in table 1 which cohort/procedure are corresponsing to each acronim. It is not clear. The list of acronyms should be produced at the end of the manuiscript as there are many and sometimes it is difficult to follow the text without a easy to reach legend.
References
1. Girardelli, S., Cavoretto, P.I., Origoni, M., Gaeta, G., Albano, L., Acerno, S., Mortini, P., Lamis, F.C., Peralta, C.F., & Candiani, M. Surgical approaches to in-utero spina bifida repair: a systematic review. Italian Journal of Gynaecology and Obstetrics; 34, 4: 2022. DOI: 10.36129/jog.2022.16
2. Botelho RD, Imada V, Rodrigues da Costa KJ, Watanabe LC, Rossi Junior R, De Salles AAF, et al. Fetal Myelomeningocele Repair through a Mini-Hysterotomy. Fetal Diagn Ther. 2017;42(1):28-34. doi: 10.1159/000449382.
3. Peralta CFA, Botelho RD, Romano ER, Imada V, Lamis F, Junior RR, et al. Fetal open spinal dysraphism repair through a mini-hysterotomy: In[1]fluence of gestational age at surgery on the peri[1]natal outcomes and postnatal shunt rates. Prenat Diagn. 2020;40(6):689-97. doi: 10.1002/pd.5675
Reviewer 2 Report
study design: 64 patients were selected for the study group (SC), but only 60 patients were qualifying for fetal surgery. Please explain
There are always mentioned 4 comparable groups of open fetal MMC repair. At a certain point another group of women who underwent fetoscopic repair is mentioned. This is confusing. I would recommend to skip this group.
Mainly, the group describes two changes in their protocol. The surgical uterine access and skipping magnesium as a tocolytic agent. It is unclear which change affected the outcome in which way. Can you clarify on that?
Round 2
Reviewer 1 Report
The article was improved after revision and tentatively accepted for publication. However I have a further concern. There is no mention of the lenght of hysterotomy: this information should be provided. In the picture it appears rather short 3-4 cm (similar to the Peralta tecquique which is 3 cm) however they talk about classical hysterotomy (the MOMS trial describes an hysterotomy of 6-8 cm). Please describe the lenght of hysterotomy. If there were hybrid cases, provide mean and standard deviation, range of the histerotomies carried out.
Author Response
Thank you for invaluable comments. Indeed, the readers might be misled about the length of the hysterotomy, especially in the image analysis of Fig. 1c. - About 3-4cm (Peralt technique). Our technique of hysterotomy - classical hysterotomy - is in line with the technique described by Moron et al., where the length of the hysterotomy consists of: the initial opening – 1 cm and the working length of the DeBakey clamp - 56 mm. The total length of the hysterotomy in each case is a total of 66mm, which is consistent with the MOMS protocol. This important detail has been clarified in the text. We modified the description of the Figure and added information about the standard length of the incision. A small 1-2 mm variability might have existed but was not documented in the OFS protocols.
Corrected Fig 1. description
Fig. 1/b. placement of the DeBakey clamps in the line of the incision and fixation of the amniotic membrane to the uterus with continuous suture. The total length of the uterine opening in complete relaxation is 66 mm (initial incision of 10 mm and final – working length between DeBakey clamp of 56 mm).
Fig. 1/c. closure of the amniotic cavity with amniotic membrane fixation to the incised edge of the uterus with continuous suture. The length of the uterine opening is only 45 mm due to increased uterine tone.
Thank you for your comments. We hope that you will approve of the changes which were made, as per your suggestions. Should you have any further comments and suggestions, we will be most grateful to receive them.
Reviewer 2 Report
I would still recommand to skip the fetoscopic part in the manuscript. Fetoscopy is not introduced in the introduction. The focus in this work is on implementing a new tocolytic regimen and describes the center's special hysterotomy technique. At one point fetoscopic data show up which have nothing to do with the rest of the manuscript.
Author Response
Thank you for your comments and invaluable suggestions. Indeed, the comparison with the fetoscopic results published by the IFMRC disrupts the continuity of the research discussion, which is not integral to the topic of our study. This part has been removed from the discussion, which will undoubtedly make the discussion more cohesive. Also, the manuscript has been proofread by a certified specialist.